# Multi-Modal Conditional GAN:
# Data Synthesis in the Medical Domain

**Jonathan D. Ziegler**
Novartis Institutes for
BioMedical Research
Basel, Switzerland

**Sajanth Subramaniam**
Novartis data42 AG
Basel, Switzerland

**Michela Azzarito**
Novartis data42 AG
Basel, Switzerland

**Orla Doyle**
Novartis Ireland Limited
Dublin, Ireland

**Peter Krusche**
Novartis Pharma AG
Novartis
Basel, Switzerland

**Thibaud Coroller**
Novartis Corporation
East Hannover, NJ, USA
`thibaud.coroller`
`@novartis.com`

## Abstract

Despite continuous collection of linked clinical and imaging datasets within the drug development process, it remains challenging to analyze those data to improve our understanding of disease and treatment. Data collection is often implemented inconsistently across studies or study sites, specific data modalities may be missing (e.g. lab measurements or medical images), and patient consent and data privacy laws constrain the purpose for which data may be used. In this paper we propose a method for conditional data generation across tabular and imaging modalities as a solution to overcome some of these challenges by generating synthetic patient data that are both realistic and complete across modalities. Our method, the multi-modal conditional GAN (MMCGAN), combines a conditional GAN for tabular data alongside a model for conditional 3D image synthesis at variable resolution. Our method brings a novel combination of capabilities: joint, scalable and efficient conditional data synthesis for clinical and full resolution 3D imaging data.

## 1 Introduction

Drug development research and healthcare systems around the world are continuously generating vast amounts of data and measurements on many aspects of disease and treatment (see [1, 2, 3] for examples of such datasets): Clinical studies collect detailed information on treatment and disease progression for new drugs, and health care providers collect data by tracking diagnostic procedures and prescriptions. For example, during clinical trials, a patient typically undergoes numerous examinations to evaluate their disease state and their response to an intervention (treatment). These data sources enable us to construct a rich database, including clinical and demographic information as well as common lab values. In addition, some trials may augment data collection by including data modalities such as genetic sequencing, proteomics, medical imaging and real-world data (e.g. as a history of interactions with the healthcare system). These data types, which are often unstructured in nature, introduce new analytical complexity alongside traditional data sources. Developing methods for integrating these data sources in a meaningful way is challenging. Furthermore, heterogeneity and inconsistency during data collection, as well as the critical requirement to respect data privacy for study participants add to the challenges of using those data in practice.

NeurIPS 2022 Workshop on Synthetic Data for Empowering ML Research.

We believe that data synthesis and imputation across modalities provides a key methodological opportunity towards addressing many of these challenges and deriving value from the complex data assets that are being collected. In this paper we present a method for multi-modal conditional data synthesis where the goal is to train generative models for linked data modalities, such as medical images for specific imaging phenotypes (here we focus on body mass index), or clinical measurements along with a set of images. A key aspect of this method is that it preserves cross-modal correlations and provides a flexible tool to study these by conditional data generation.

## 2 Methods

A wide range of methods exists for generating synthetic medical data. The most suitable method should take a number of factors into account: (i) data type (for example images, text, tabular), (ii) data dimensionality, (iii) data distribution and sparsity (e.g. number of modes), and finally (iv) the number of examples available for training. Clinical data (typically structured data in tabular form) have been successfully generated by means of distribution matching and sampling, as effectively demonstrated by Patki et al. [4] and revisited by Kamthe et al. [5]. Deep learning assisted methods using generative adversarial networks [6] in combination with distribution pre- and post-processing were introduced as the Conditional Tabular GAN (CTGAN) architecture shortly after by Xu et al. [7]. Using nonlinear synthesis methods such as GANs provides greater modeling capacity than distribution matching and sampling, and is more likely to match the original multi-dimensional distribution of the data, even for complex datasets. Image generation is an active research area, however, medical imaging typically requires highly specific solutions depending on the imaging technology [8, 9, 10, 11]. Generative Adversarial Networks (GANs) producing photo-realistic images, such as StyleGAN2, have received widespread coverage in popular media [12]. More recently, diffusion models have significantly advanced the state of the art and resulted in an early adaptation of generative modelling for artistic image generation [13, 14, 15]. While these methods have shown success on 2D images, only very few have been shown to work in practice on 3D medical images Pinaya et al. [16]. Adapting modern generative models designed for two-dimensional image synthesis to the three-dimensional nature of medical images is a non-trivial task. Memory and compute limitations of current deep-learning hardware make direct adaptation prohibitively costly. Multiple approaches have been investigated to tackle this challenge. The $\alpha$-GAN model proposed by Kwon et al. [17] directly produces synthetic images of $64^3$ voxels, larger image sizes are created by means of upsampling (artificially increasing image resolution using interpolation without creating true higher resolution). Uzunova et al. [18] combine the generation of low resolution images of size $64^3$ with subsequent upsampling-GANs that take smaller patches of the generated low-resolution image as an input and produce high-resolution patches, resulting in an image size of up to $512^3$ voxels. The hierarchical amortized method described by Sun et al. [19] overcomes these limitations with the proposed HA-GAN model architecture: their proposed model splits the synthesis task into two branches. One branch focuses on synthesizing global images at low resolution ($64^3$ voxels), while the second branch produces partial full-resolution images (sampled sub-volumes). Both branches feed into a shared core generator and are combined with their respective discriminator modules. This approach has the key advantage of jointly training the model to retain global structures and local details without the necessity of processing whole high-resolution images, thus significantly reducing memory requirements.

To model data for medical applications, it is key to maintain the structure across multiple modalities, such as 3D images and tabular data to ensure the utility of such datasets. While multimodal image synthesis has been investigated for neuroimaging applications by Lan et al. [20], at the time of this paper, no substantial work could be found on the simultaneous synthesis of multiple data types.

### 2.1 Multi-head Generative Adversarial Networks

In this paper we introduce our novel method for the conditional synthesis of multi-modal data with applications in the medical domain. This core functionality is achieved using a multi-head generative adversarial network called the multi-modal conditional GAN (MMCGAN). Classical GANs use one generator and one discriminator, trained in an adversarial manner. While training the generator module iteratively improves the fidelity and diversity of the generated data, the discriminator's improving detection capability is used to detect remaining weaknesses in the generated data, subsequently improving the generator in the next iteration. Our proposed method uses a multi-head GAN architecture built around a shared generator core. The output of this shared generator is a latent

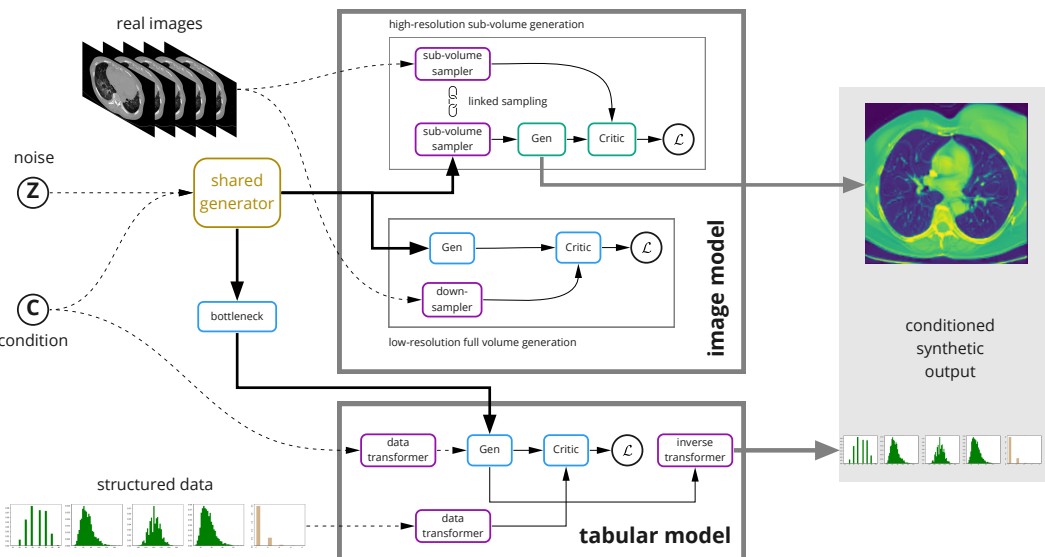

Figure 1: The Multi-Modal Conditional GAN (MMCGAN) architecture uses a shared generator to transform the input vector consisting of noise and conditions into a latent hypercube, which is subsequently transformed into multimodal outputs by the individual heads of the network, preserving cross-modal correlations.

hypercube of size $64^4$, subsequently fed into the individual generator heads, where each head is responsible for producing data of a specific modality. For the tabular synthesis head, the hypercube is reduced through a bottleneck module and passed through a CTGAN core model. This creates a numerical representation of the defined clinical covariates, which are transformed into human-readable form by the transformations discussed in Section 2.3. Image synthesis is implemented by combining two convolutional heads into the hierarchical amortized configuration described in section 2.2, which, at inference time, outputs images of $256^3$ voxels. Cross-modal correlation is encouraged by back-propagating the gradients of each output head to the shared latent generator, which contains the majority of the system's modeling capacity. This joint training of a shared generator across modalities and a subsequent processing of it's output for specific data types presents a key benefit of the proposed method: while each individual output head is only sensitive to a specific data modality, the joint back-propagation ensures that the data remains consistent across all modalities. A high-level network architecture and signal flow can be found in Figure 1. The detailed structure of the individual components can be found in the supplementary material in Figures A.1 and A.2.

## 2.2 High-resolution image synthesis

The described multi-head model architecture matches the shared generator architecture of the HA-GAN model family, which therefore can easily be adapted for the image synthesis head. In the proposed case, two heads of the MMCGAN model can be combined to represent an HA-GAN model. As outlined in Section 2, one head creates low-resolution, full-volume images of $64^3$ voxels, while the other produces high-resolution image sub-volumes of $256 \times 256 \times 32$ voxels. The method uses sampling during training to extract a sub-volume of the shared latent hypercube which is passed to the high-resolution image generation head. Additionally, the real image input for the associated discriminator is cropped to a sub-volume of correct dimension. At inference time, when gradient calculation is no longer necessary, the low-resolution head and all discriminators can be disregarded. This significantly relaxes the memory constraints and the model is capable of directly generating three-dimensional images of $256\,\text{mm} \times 256\,\text{mm} \times 256\,\text{mm}$ size with a $1\,\text{mm}$ resolution without any additional stitching or upsampling. Technically, this is achieved by removing the sub-sampler from the generation path.

## 2.3 Tabular data handling

Structured data such as clinical patient information can require elaborate preprocessing for use with a deep learning model. The Synthetic Data Vault (SDV) provides all transformations required to convert a structured table into a numerical matrix that can be ingested by the MMCGAN network [4] and provides an excellent pipeline for this complex task. Additionally, the Conditional Tabular GAN (CTGAN) proposes a generative model architecture and data ingestion for multimodal tabular data synthesis [7]. Normalization of numerical columns is performed in a mode-specific manner: columns are approximated using a Variational Gaussian Mixture Model (VGM) [21] and the numerical value is expressed as a one-hot vector representing the mode number, which has been sampled according to the probability distribution for the given value, and the value associated with the given mode. This data transformation and the CTGAN's core model are adapted to match the MMCGAN multi-head architecture and data pipeline and are used to create the tabular data head.

## 2.4 Conditional data synthesis

Manipulating the output of a GAN by means of a conditioning is a commonly used technique [22]. Generally, a vector describing the desired point in the condition space is concatenated with the random noise vector passed to the generator. An additional auxiliary classifier can be added to the discriminator. This expands the model's loss function to contain a (multi-class) classification loss [23]. For the proposed model, a multi-conditional input vector is passed to the shared generator. Additionally, the conditions are transformed for the tabular head according to the transformations described above and added to the input of the tabular head.

## 2.5 Training data and process

All results shown in the following sections were generated based on a training dataset containing 1018 patients, each providing 22 clinical columns and $\geq 1$ three-dimensional lung CT images of $256\,\text{mm}^3$ size at $1\,\text{mm}$ resolution. Pytorch was used to define the model architectures and required data processing and training pipelines. All models were trained for 200 epochs using default Adam optimizers with the optimization parameters shown in Table 1 of the supplementary material. The image heads were updated every $5^{th}$ iteration. This was done to compensate the relatively low count of update steps when compared to the convergence pattern of a CTGAN baseline model. All personal health information (PHI) were removed or encrypted by a proprietary, automated de-identification engine (i.e., prior to receipt) obviating the requirement for informed consent.

## 3 Results

A fully synthetic surrogate dataset of 10000 data points, consisting of 3D images and clinical covariates, was generated. The images are $256\,\text{mm}^3$ large with a resolution of $1\,\text{mm}$ and show synthetic three-dimensional chest CT images. The clinical data included 22 covariates, with demographic, treatment and follow-up information. It consists of 7 categorical, 9 binary, 2 integer, and 4 float covariates of unknown and non-normalized distributions. The synthesized dataset can be used to facilitate the development of downstream analysis and evaluation pipelines without exposing the real target dataset during development. The quality of synthetic images was verified both visually (see Figure 4) and by means of low-dimensional visualization. In order to create two-dimensional feature maps of real and synthetic images, the backbone of a pretrained Medicalnet3D network as proposed by Chen et al. [24] was used for feature extraction. The extracted feature vectors were reduced to a dimensionality of 2 by training a UMAP model [25] on the real dataset, and applying this to both the real and synthetic data vectors. Figure 2 shows two-dimensional representations of the extracted projections for real and synthetic images and clinical data. Both diversity and fidelity can be observed in the UMAP with respect to outliers in the synthetic data and underrepresented areas in the real data. Additionally, the class cutoffs can be visually verified: a significant overlap of synthetic samples with matching labels in the real data can be observed. This clearly disproves a null hypothesis of randomly distributed class labels over the entire UMAP.

Quality control for the tabular data was performed using both pre-built evaluation toolkits from the SDV and table-evaluator packages [26], and by means of two-dimensional distribution analysis by way of UMAP reductions. Univariate distributions can be compared between real and synthetic tables

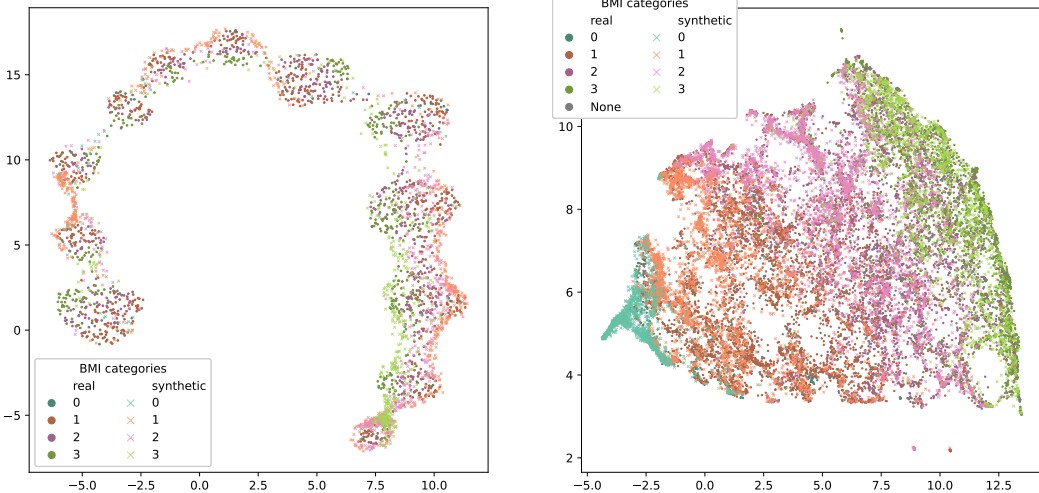

Figure 2: UMAP feature visualization of real and synthetic image data in two dimensions. Clinical data (left) and images (right) are both shown conditioned on BMI.

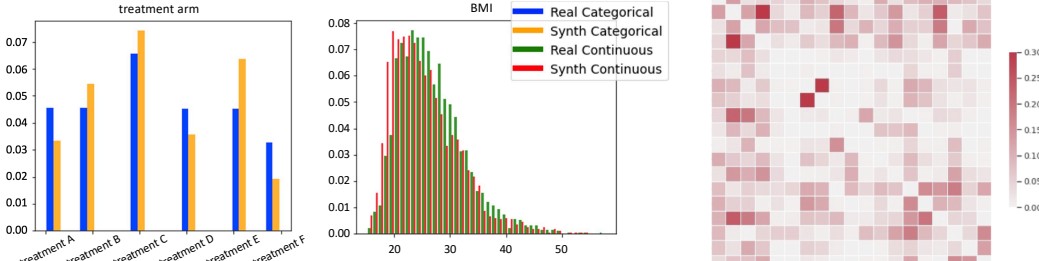

(a) Example Comparison of univariate distributions between real and synthetic data. Both continuous (red/green) and discrete (tan/blue) columns are modeled. The examples show the discrete treatment arm (left) and continuous BMI (right) covariates.

(b) Difference in cross correlation between real and synthetic clinical covariates. See supplementary Figure A.5 for details.

Figure 3: Univariate and bivariate comparison of real and synthetic data. Both discrete and continuous distributions are well represented. As seen in Fig. 3b, existing bivariate correlations are maintained in the synthetic dataset.

as a first line of quality control of the synthesis methods. Additionally, bivarate correlation analyses provide some insight into the retention of complex cross-dependencies in the synthetic data.

## 3.1 Medical data synthesis conditioned on body mass index

The Body Mass Index (BMI), calculated as $^{\text{weight}}/_{\text{height}^2} \, \text{kg} \, \text{m}^{-2}$, can be used as a principal proof of concept for conditional synthesis, as the correlation can easily be verified visually. Figure 4 shows the synthesized high-resolution images corresponding to four points in the $\mathbb{R}^{1024}$ latent space. For each virtual patient, the BMI is conditioned, as seen on the vertical axis, in categories 0 through 3:

$$\text{BMI}_0 \leq 18.5 \, \text{kg} \, \text{m}^{-2} < \text{BMI}_1 \leq 25.0 \, \text{kg} \, \text{m}^{-2} < \text{BMI}_2 \leq 30.0 \, \text{kg} \, \text{m}^{-2} < \text{BMI}_3$$

The increase in low-density fatty tissue is clearly observable for the randomly generated samples. This behaviour was statistically verified by observing real and synthetic datasets with regard to the correlation between BMI and total occupied volume of matter with a density in the range of $-150 \, \text{HU}$ to $300 \, \text{HU}$. This correlation is statistically compared with that of real data with known BMI. As seen in Figure A.4 in the supplementary material, both real and synthetic datsts show a clear correlation between the image data and the associated BMI category.

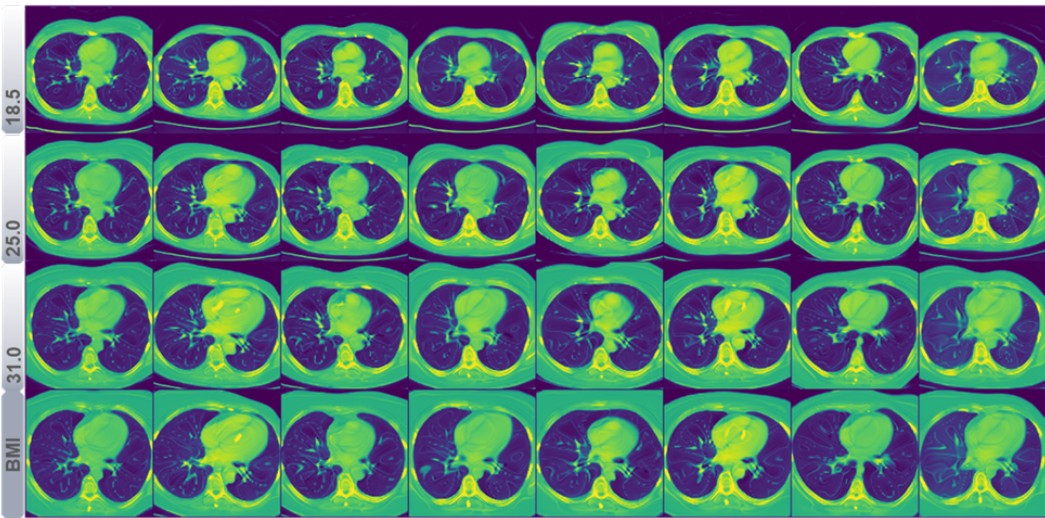

Figure 4: Transverse center slice through randomly generated three-dimensional images under varying BMI conditioning. Eight points in the latent sampling space are converted to data, as shown on the horizontal axis. On the vertical axis, the conditions accompanying the noise input are varied, with BMI increasing from top to bottom.

# 4 Discussion

We did not explicitly focus on the privacy aspects of synthetic data in this paper, although we believe that extending our models to address disclosure risks is possible. The standard approach to privacy used in literature is the differential privacy model by Dwork [27], and a wealth of literature has been developed describing how models can be trained with guarantees of differential privacy, see e.g. [28] for a very generic method to implement differential privacy in deep learning. Moreover, requirements for data handling are typically regulated through privacy laws and patient consent, see e.g. El Emam et al. [29] for an overview and discussion on practical privacy with synthetic data. We believe that our methods can be extended to include privacy guarantees. In this work, we focus on multi-modal data generation and ensure privacy by removing all personal health information from our clinical data using standard commercially available methods (see Mallon et al. [1] for a discussion of data preprocessing in multi-modal clinical trial datasets).

When using comparatively simple conditions, such as BMI, our models produce results that show a high level of diversity and fidelity, and match the original multimodal distribution well. From a biological perspective, the conditioning results match the expected behavior, as seen in Figure A.4 in the supplementary material. This already enables a first application of our method: Testing and development of machine learning pipelines. For example, we can randomize labels during data synthesis and supply realistic-looking training data without compromising the results of the actual machine learning experiment while testing. For scientific applications, data synthesis must be able to handle more complex conditions. When using more complex, multivariate features, some clear differences between the image categories can be observed, although medical relevance and biological accuracy need further verification. This can be seen as an initial step into gaining scientific insights through data synthesis: We can explore how clinical features are linked to images through conditional data synthesis. In the setting of clinical studies, where we typically study only thousands of subjects (rather than millions as e.g. in real-world cohorts / biobanks), we believe this is a useful method to assess if prediction of outcomes from images is a useful direction, or if spurious features in the images are being modelled. Finally, the limitations of cross-modal imputation require additional investigation: For example, with better understanding of the biases introduced through this method, it could be used to generate augmented hybrid datasets which have a complete set of measurements and data modalities available for all subjects. These datasets could then be used to increase the effective sample size when training models, as well as handling imbalanced training scenarios.

# 5 Conclusion

This paper highlights many challenges that arise when dealing with medical data, such as inconsistent or heterogeneous data availability across modalities, and patient privacy. The authors present a novel method for conditional medical data synthesis which approaches the previously unexplored challenge of correlated synthesis across multiple modalities, in particular three-dimensional images and structured tabular data. The method was used to create a synthetic dataset containing 10000 data points consisting of high-resolution, 3D chest CT images and 22 associated clinical covariates. The Body Mass Index was chosen as the conditioning variable. The modular network architecture and end-to-end training process ensure a simple transition to other use-cases and data modalities. Additionally, using more complex multivariate conditions are a promising path toward the generation of medical insights. Both aspects are active research topics which the authors wish to expand on in their future work.

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

# A Supplementary material

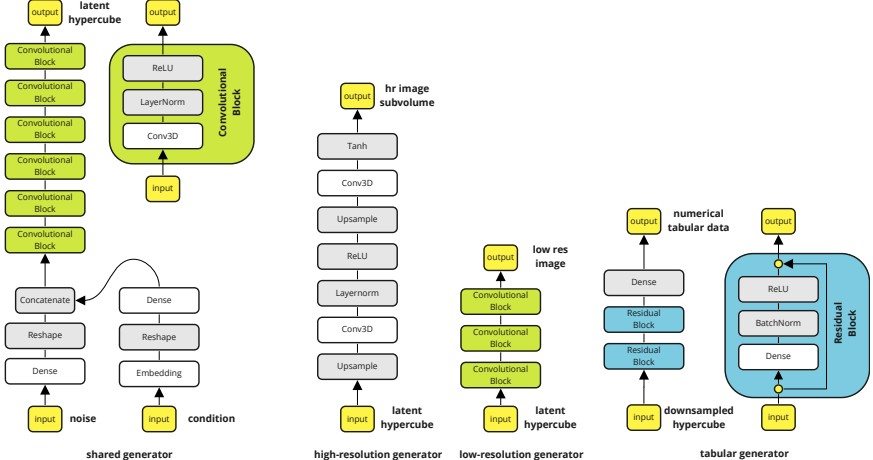

Figure A.1: All generators used in the MMCGAN architecture. While the image heads and most of the shared generator are fully convolutional, the tabular generator uses residual blocks consisting of fully connected layers.

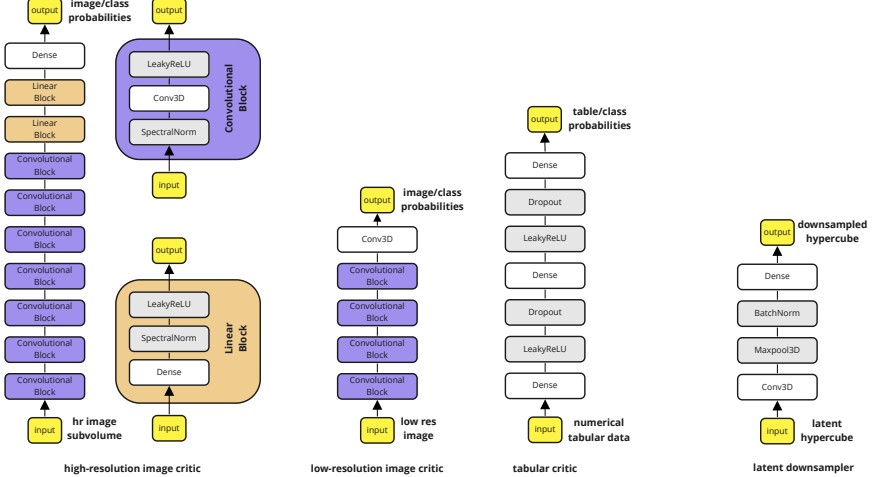

Figure A.2: All critics and the downsampling module used in the MMCGAN architecture.

Table 1: Optimization parameters for individual model components in MMCGAN architecture.

| model component | learning rate | $\beta_1$ | $\beta_1$ |
|---|---|---|---|
| shared generator | $1 \times 10^{-4}$ | 0 | 0.999 |
| image generators | $1 \times 10^{-4}$ | 0 | 0.999 |
| tabular generator | $2 \times 10^{-4}$ | 0.5 | 0.9 |
| image discriminators | $4 \times 10^{-4}$ | 0 | 0.999 |
| tabular discriminator | $2 \times 10^{-4}$ | 0.5 | 0.9 |
| tabular bottleneck | $2 \times 10^{-4}$ | 0.5 | 0.9 |

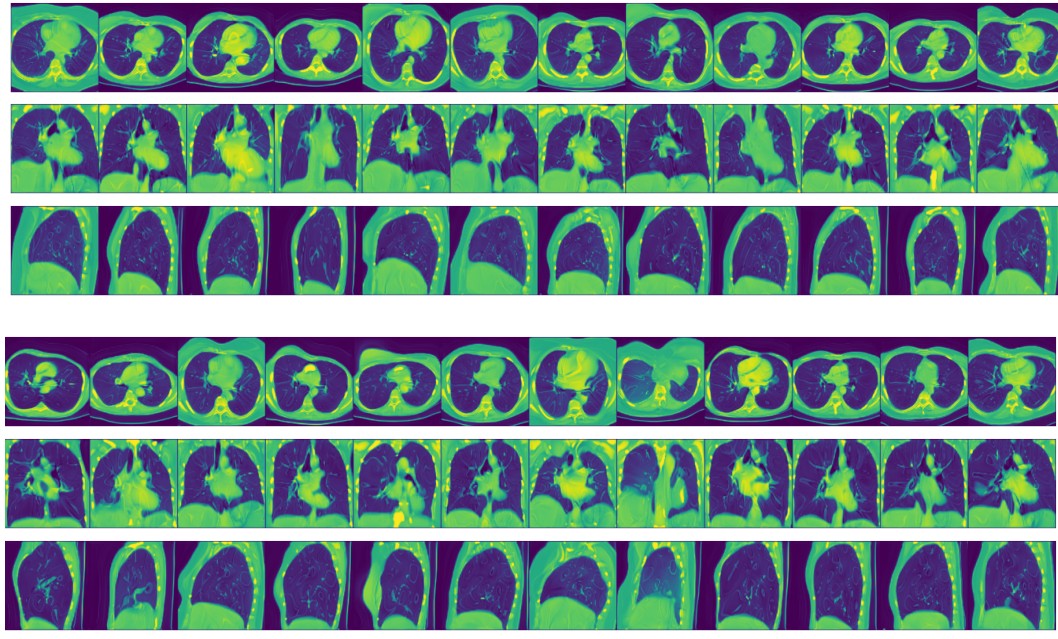

Figure A.3: Random image samples generated using the MMCGAN model. Horizontal axis shows $2 \times 12$ different virtual patients, vertical axis shows slices through main axes.

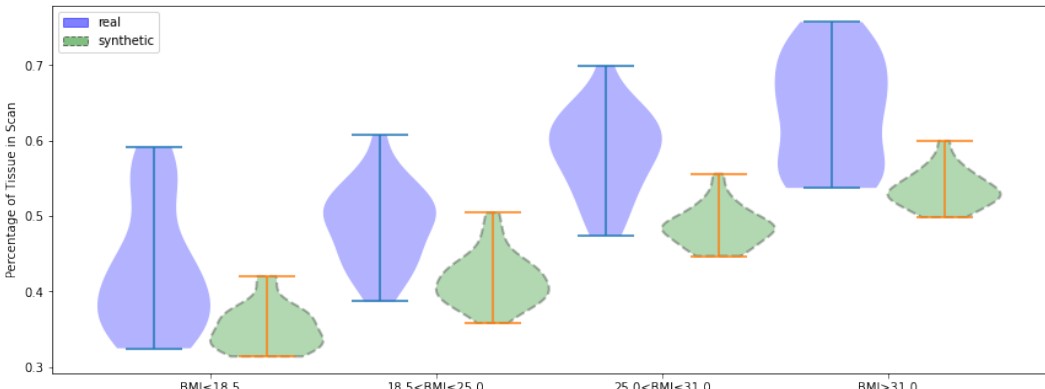

Figure A.4: Real and synthetic data are compared with respect to percentage of scan occupied by fat, tissue, and water, compared to Body Mass Index of subjects.

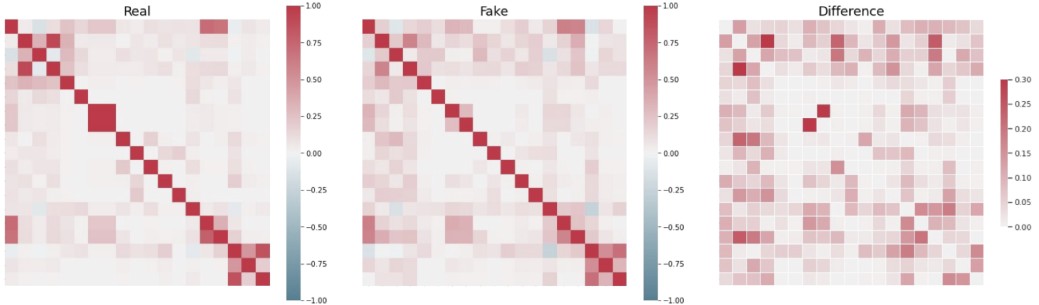

Figure A.5: Comparison of bivariate cross-correlation between real (left) and synthetic (middle) data. The difference signal is shown in the rightmost image. Much of the data set's correlation is retained in the synthetic surrogate.

