# OpenReview forum: "Multi-Modal Conditional GAN: Data Synthesis in the Medical Domain"
_NeurIPS.cc/2022/Workshop/SyntheticData4ML — Neurips 2022 SyntheticData4ML_

### Official Review · Reviewer_xzDH · 2022-10-16
**Authors presents multimodal conditional GAN (MMCGAN) method to syntethis tabular data and  3D chest CT images. MMCGAN method has the potential to synthesise clinical and full resolution 3D image data.**

**Rating:** 6
**Confidence:** 3

**Review:**

## Originality
Authors claim to present a novel method Multi-modal conditional GAN, stating that no substantial work could be found, to synthesise multiple data types of 3D chest CT images and Body Mass Index.


## Significance
This work is of importance significance to the workshop as present results on synthesising BMI and 3D CT volumetric data that might lead to interesting discussions on the use of other tabular data as well as future work on increasing sample size to train models with imbalanced  datasets.

## Clarity
Clarify of the paper require more work as :

* The current work do not properly cite works when referring to CTGAN the first time in the paper. That lead me to think that authors appear might like to do further literature review as there are some body of work on  “CTGAN multimodal” https://scholar.google.com/scholar?hl=en&as_sdt=0%2C5&q=CTGAN+multimodal&oq=CTGAN+ and “multimodal GAN medical” https://scholar.google.com/scholar?hl=en&as_sdt=0%2C5&q=multimodal+GAN+medical&btnG=

Also authors might like to check the latest citations of the following works:

Darabi, Sajad, and Yotam Elor. "Synthesising Multi-Modal Minority Samples for Tabular Data." arXiv preprint arXiv:2105.08204 (2021). See citations: https://scholar.google.com/scholar?cites=6431312146576520169&as_sdt=2005&sciodt=0,5&hl=en

Xu, Lei, Maria Skoularidou, Alfredo Cuesta-Infante, and Kalyan Veeramachaneni. "Modeling tabular data using conditional gan." Advances in Neural Information Processing Systems 32 (2019). See citations: https://scholar.google.com/scholar?cites=3578506996923518478&as_sdt=2005&sciodt=0,5&hl=en





## Miscellaneous

ABSTRACT
* Authors might like to be more specific on the resolution of the images.
* Authors mentions that their method is scaleable and efficient but not sure if such description are somewhat subjective. Perhaps authors would like to add further results on how their method was scalable (production ML pipeline) and efficient (?).

INTRODUCTION
* Not sure if authors only present anecdotal experiences in the first paragraph. I suggest adding further seminal papers to have better evidence of previous works.
* In the second paragraph, it seems that authors talk about privacy challenges and future work that can be moved to the conclusions section or perhaps added as appendix.
* Can authors cite those “commercially available methods” and provide detailed description of such methods?

METHODS
* Can authors talk about more about those specific solutions?: “medical imaging typically requires highly specific solutions depending on the imaging technology” Perhaps authors would like to cite some seminal papers here?

* Is CTGAN referring to Colour Translation GAN?, I think authors would like to add the reference on CTGAN.

* How the images were verified? Can authors add further details on such verification? Do authors used Turing visual test or other metrics to verify synthesised images?
> “The quality of synthetic images was verified both visually (see
Figure 4) and by means of low-dimensional visualization”.
I think the paper requires more work to be more clear on how these synthesised was verified as this comes in the result section with the univariate and bivariate comparisons.


RESULTS

* Can authors talk more on how the class cut-offs is visually verified?
> “Additionally, the class cutoffs can be visually verified.”
* Authors refer to the supplementary material for further relevant sutures on statistical correlation which might be much better results to be present in the main paper.



OTHERS

* Can authors use CT images with greyscale for figure 1 and 4?
* Authors can incorporate supplementary material as appendix.
* What are the units if Figure2? Not quite sure if circles and crosses are well differentiated in the figure, perhaps authors would like to use different colours, augment the size of the symbols, or zooming out to show much better such results?
* Perhaps, authors would like to talk about the areas where there is not good matches and the reason of why that happen in Fig2?

---

### Official Review · Reviewer_2gWT · 2022-10-18
**multimodal conditional GAN for medical data generation, incl 3D imaging.**

**Rating:** 5
**Confidence:** 4

**Review:**

The paper proposes a multimodal conditional GAN for medical data generation, incl 3D imaging.
The topic is relevant and timely. The paper is well-written and easy to follow.

The GAN components of the GAN architecture are not quite novel, but the application is interesting and the model combined existing components in an interesting way to achieve multimodal conditional GAN.

While some discussion of the state-of-the-art is provided, no experimental comparison is performed.
In the experiments, conditioning is performed on a single ordinal variable only representing BMI. This seems like a fairly limited setting.

The individually generated data modalities and cross correlation between clinical variables are evaluated in a purely qualititatve manner using a 2D UMAP and visually showing a correlation matrix. Also images are only qualitatively compared. It is unclear, if images represent the conditional BMI groups well.

While the model is motivated by applications in data synthesis and imputation, the current paper only focuses on data synthesis. The topic of data imputation is not studied in this paper. Given that the model also only conditions on a single ordinal variable, it seems as if significant further changes to the model would be needed.

In sum, interesting model and application. Yet, the experiments are limited as no comparison is provided, only a single ordinal variable is used for conditioning and all results are qualitative in nature.

---

### Meta-Review · Area_Chair_PzPQ · 2022-10-19

**Recommendation:** Accept

**Review:**

Highly relevant problem and interesting approach, but lacking thorough empirical evaluation. I hope the authors will incorporate the reviewer's feedback regarding additional baselines, quantitative metrics, and datasets to further improve the paper.